

# Identifying bias in network clustering quality metrics

Martí Renedo-Mirambell and Argimiro Arratia

Soft Computing Research Group (SOCO) at Intelligent Data Science and Artificial Intelligence Research Center, Department of Computer Sciences, Universitat Politécnica de Catalunya, Barcelona, Spain

## ABSTRACT

We study potential biases of popular network clustering quality metrics, such as those based on the dichotomy between internal and external connectivity. We propose a method that uses both stochastic and preferential attachment block models construction to generate networks with preset community structures, and Poisson or scale-free degree distribution, to which quality metrics will be applied. These models also allow us to generate multi-level structures of varying strength, which will show if metrics favour partitions into a larger or smaller number of clusters. Additionally, we propose another quality metric, the density ratio. We observed that most of the studied metrics tend to favour partitions into a smaller number of big clusters, even when their relative internal and external connectivity are the same. The metrics found to be less biased are modularity and density ratio.

## INTRODUCTION

Clustering of networks is a very active research field, and a wide variety of clustering algorithms have been proposed over the years (*e.g.*, a good survey is found in *Fortunato & Hric, 2016*). However, determining how meaningful the resulting clusters are can often be difficult, as well as choosing which clustering algorithm better suits a particular network. In many cases, various clustering algorithms can give substantially different results when applied to the same network. This is not only due to the limitations or particularities of the algorithms, but also to some networks possibly having multiple coexisting community structures.

Our goal is to study how existing cluster or community quality metrics behave when comparing several partitions of the same network, and determine whether they properly show which of them better match the properties of a good clustering, or if they are biased in favour of either finer or coarser partitions. Knowing this is essential, for there could be cases in which a cluster quality metric simply scores better than another because it tends to find smaller or larger clusters, with less regard for other properties, and not because it is better at revealing the structure of the network.

We selected a few popular cluster quality metrics, both local, which assign a score to evaluate each individual cluster (*e.g.*, conductance, expansion, cut ratio, and others), and global, which evaluate the partition as a whole (*e.g.*, modularity). We also propose a new

Corresponding author
Argimiro Arratia,
argimiro@cs.upc.edu

metric, the density ratio, which combines some of the ideas behind other metrics, in an attempt to improve over some of their limitations, and more particularly, to avoid bias caused by the number of clusters of the partitions.

Our analysis is split in two parts. In the first part, we use the stochastic block models (*Holland, Laskey & Leinhardt, 1983*; *Wang & Wong, 1987*) to generate networks of predetermined community structures, and we then compute the correlation of each metric to the size and number of communities. On the second part, we define networks with a two level hierarchical community structure (with the lower partition being a refinement of the upper partition), where one additional parameter controls the strength of one level respect to the other. Then, by varying this parameter and evaluating the quality metrics on both levels, we can see if certain metrics are biased towards finer or coarser partitions. These networks with multi-level community structure are implemented on two different models that include communities: a stochastic block model and a preferential attachment model which results in networks with a scale-free degree distribution.

## Related work

We briefly survey some of the studies in cluster quality metrics and analyses of their performance which are relevant to this work. A milestone is the work by *Yang & Leskovec (2015)*, where they analysed and classified many popular cluster scoring functions based on combinations of the notions of internal and external connectivity. More recently, *Emmons et al. (2016)* study the performance of three quality metrics (modularity, conductance and coverage), as well as that of various clustering algorithms by applying them to several well known benchmark graphs. For the case of modularity, it was already shown in *Fortunato & Barthélemy (2007)* to have a resolution limit below which small and strong communities are merged together, even when that goes against the intuition of what a proper clustering should be. This limit depends on the total amount of edges in the network, in such a way that it is more pronounced the larger the network and the smaller the community. *Almeida et al. (2011)* do a descriptive comparison of the behaviour of five cluster quality metrics (modularity, silhouette, conductance, coverage, and performance) for four different clustering algorithms applied over different real networks, to conclude that none of those quality metrics represents the characteristics of a well-formed cluster with a good degree of precision. *Chakraborty et al. (2017)* survey several popular metrics, comparing their application to networks with ground truth communities to the results of a selection of clustering algorithms, though the potential of bias relative to cluster size is not addressed.

Hierarchical or multilevel stochastic block models have been mostly used for community detection by trying to fit them to any given graph. A good example of this technique is *Peixoto (2014)*; further applications of multilevel SBM and variants for the problem of community detection are surveyed in *Lee & Wilkinson (2019)*, *Paul & Chen (2016)*, *Funke & Becker (2019)*, and more recently the works (*Peel & Schaub, 2020*; *Mangold & Roth, 2023*) study the limitations of multilevel SBM for detecting a planted hierarchy of partitions in a network. Here we use the SBM and multilevel SBM for the purpose of generating networks with predetermined community structure. Of utmost importance is the Barabasi-Albert preferential attachment model with community

structure construction of *Hajek & Sankagiri (2019)*, which motivates our own construction of a multilevel block model with preferential attachment to produce networks with community structure and degree distribution ruled by a power law. The proof of this latter fact is a novel contribution of this article.

There is scarce literature in the use of SBM and multilevel SBM, let alone the preferential attachment model for constructing synthetic ground truth communities in networks as benchmarks for testing clustering algorithms. A notable exception is *Peel, Larremore & Clauset (2017)*, that uses SBM in this sense to explore how metadata relate to the structure of the network when the metadata only correlate weakly with the identified communities. This article contributes to the literature of clustering assessment through probabilistic generative models of communities in networks.

## METHODS

In this section we detail our methodology. Portions of this text were previously published as part of a preprint (*Renedo-Mirambell & Arratia, 2021*).

### Cluster quality metrics

For the definitions of quality metrics we will use the following notation. Given a graph $G = (V, E)$, $n$ will denote its order (number of vertices) and $m$ its size (number of edges). Following common practice we may identify the set of vertices $V$ with the initial segment of positive integers $[n]$. Similarly, $n_S$ and $m_S$ will be the order and size of a subgraph $S$ of $G$, and $c_S$ will be the number of edges of $G$ connecting $S$ to $G \backslash S$. Here, in the context of community detection, we will only work with subgraphs induced by sets of nodes (*i.e.*, communities).

Given $P$ a partition of $G$ and $u, v \in V$, $\delta_P(u, v)$ will take value 1 if $u$ and $v$ are in the same part of $P$ and 0 otherwise. $k_v$ will denote the degree of a vertex $v \in V$, and $k_{med}$ the median degree.

We study two different kinds of quality metrics: cluster-wise (or local), which evaluate each cluster separately, and global, which give a score to the entire network. Additionally, for each local metric, we also consider the global metric obtained by computing its weighted mean, with weights being the corresponding sizes of the clusters.

We consider a collection of local metrics (or community scoring functions) introduced in *Yang & Leskovec (2015)* which combine the notions of strong internal and weak external connectivity that are expected of good clusters. Definitions of these local clustering quality metrics are summarised in Table 1. While many of them are too focused on a single property to be able to give a general overview on their own (like internal density, average degree, cut ratio, etc.), they all capture properties that are considered desirable in a proper clustering (which essentially come down to a combination of strong internal and weak external connectivity). The ones that actually combine both internal and external connectivity are the conductance and normalized cut (which happen to be highly correlated, as seen in *Yang & Leskovec, 2015*), so we will mostly focus our analysis on those.

As for global metrics, we consider modularity (*Newman, 2006*) and coverage (*Emmons et al., 2016*). Additionally, we propose another metric, which we named *density ratio*, and

**Table 1 Local scoring functions of a community $S$ of the graph $G = (V, E)$.**

| | |
|---|---|
| ↑ Internal density | $\frac{m_S}{n_S(n_S-1)/2}$ |
| ↑ Edges inside | $m_S$ |
| ↑ Fraction over median degree | $\frac{\|\{u:u\in S,\|\{(u,v)_v\in S\}\| > k_{med}\}\|}{n_S}$ |
| ↑ Triangle participation ratio | $\frac{\|\{u:u\in S,\{(v,w):v,w\in S,(u,v)\in E,(u,w)\in E,(v,w)\in E\}\neq\varnothing\}\|}{n_S}$ |
| ↑ Average degree | $\frac{2m_S}{n_S}$ |
| ↓ Expansion | $\frac{c_s}{n_s}$ |
| ↓ Cut ratio | $\frac{c_s}{n_s(n-n_s)}$ |
| ↓ Conductance | $\frac{c_s}{2m_s+c_s}$ |
| ↓ Normalized cut | $\frac{c_s}{2m_s+c_s} + \frac{c_s}{2(m-m_s)+c_s}$ |
| ↓ Maximum ODF | $\max_{u\in S} \frac{\|\{(u,v)\in E:v\notin S\}\|}{k_u}$ |
| ↓ Average ODF | $\frac{1}{n_s}\sum_{u\in S} \frac{\|\{(u,v)\in E:v\notin S\}\|}{k_u}$ |
| ↑ Local density ratio | $1 - \frac{c_s/(n_S(n-n_S))}{m_S/(n_S(n_S-1))}$ |

**Note:**
Arrows indicate whether the score takes higher values when the cluster is stronger and lower values when it is weaker (↑), or *vice versa* (↓).

**Table 2 Global scoring functions of a partition $P$ of the graph $G = (V, E)$.**

| | |
|---|---|
| ↑ Modularity | $\frac{1}{2m}\sum_{u,v}\left(A_{uv} - \frac{k_u k_v}{2m}\right)\delta_P(u,v)$ |
| ↑ Coverage | $\frac{\sum_{u,v} A_{uv}\delta_P(u,v)}{\sum_{u,v} A_{uv}}$ |
| ↑ Global density ratio | $1 - \frac{\|\{(u,v)\in E:\delta_P(u,v)=0\}\|/\|\{u,v\in V:\delta_P(u,v)=0\}\|}{\|\{(u,v)\in E:\delta_P(u,v)=1\}\|/\|\{u,v\in V:\delta_P(u,v)=1\}\|}$ |

is defined as $1 - \frac{\text{external density}}{\text{internal density}}$. It is based on some of the metrics in Table 1, but defined globally over the whole partition (see Table 2). It takes values on $(-\infty, 1]$, with 1 representing the strongest partition, with only internal connectivity, while poor clusterings with similar internal and external connectivity have values around 0. Only clusterings with higher external than internal connectivity (so worse on average than clustering randomly) will have negative values, and if we keep decreasing the internal connectivity, the density ratio will tend to $-\infty$ as the internal density approaches 0. The density ratio can be computed on linear time over the number of edges. A local version of this metric is defined in Table 1.

## Stochastic block model (SBM)

A potential benchmark for clustering algorithm evaluation is the family of graphs with a pre-determined community structure generated by the *l-partition* model (*Condon & Karp, 2001*; *Girvan & Newman, 2002*; *Fortunato, 2010*). It is essentially a block-based extension of the well known Erdös-Renyi model, with $l$ blocks of $g$ vertices, and with probabilities $p_{in}$ and $p_{out}$ of having edges within the same block and between different blocks respectively.

The generalization of this idea is the stochastic block model, which allows blocks to have different sizes, as well as setting distinct edge probabilities for edges between each pair of

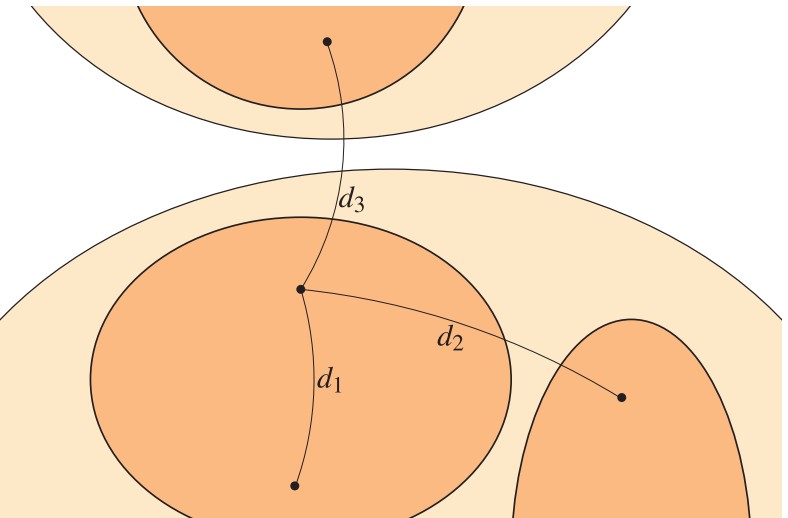

**Figure 1  Multi-level stochastic block model.** 

blocks, and for internal edges within each block (*Holland, Laskey & Leinhardt, 1983*; *Wang & Wong, 1987*). These probabilities are commonly expressed in matrix form, with probability matrix $P$ where $P_{ij}$ is the probability of having an edge between each pair of vertices from blocks $i$ and $j$. Note that this matrix is symmetric for undirected graphs. Then, graphs generated by probability matrices where the values in the diagonal are larger than the rest, will have very strong and significant clusters, while more uniform matrices will produce similarly uniform (and therefore poorly clustered) graphs.

## Multi-level stochastic block model
To generate networks with multi-level community structures, we propose a variation of the stochastic block model, defined as follows:

- $C_1,\ldots, C_n$ are the first level of communities.
- Each community is split into $C_{i_1}, \ldots, C_{i_{m_i}}$ sub-communities.
- $d_1$ is the edge probability within sub-communities.
- $d_2$ is the edge probability within communities (but with different sub-communities).
- $d_3$ is the edge probability outside communities.

Consequently, the model takes as parameters the upper and lower level block size lists, as well as the edge probabilities $d_1, d_2, d_3$. Note that the lower level partition $P_l = \{C_{i_j}\}$ is a refinement of the upper level partition $P_u = \{C_i\}$. A representation of this multi-level stochastic block model is shown in Fig. 1, where the upper level is the light coloured region whilst the lower level is the darker region, and the edge probabilities are clearly identified.

The resulting model can itself be expressed as a standard (single-level) stochastic block model, using the lower-level communities as blocks, and with probability matrix as seen in Fig. 2, which means it is actually a particular case of the standard stochastic block model. This idea can be extended to define stochastic block models with a hierarchical or multi-

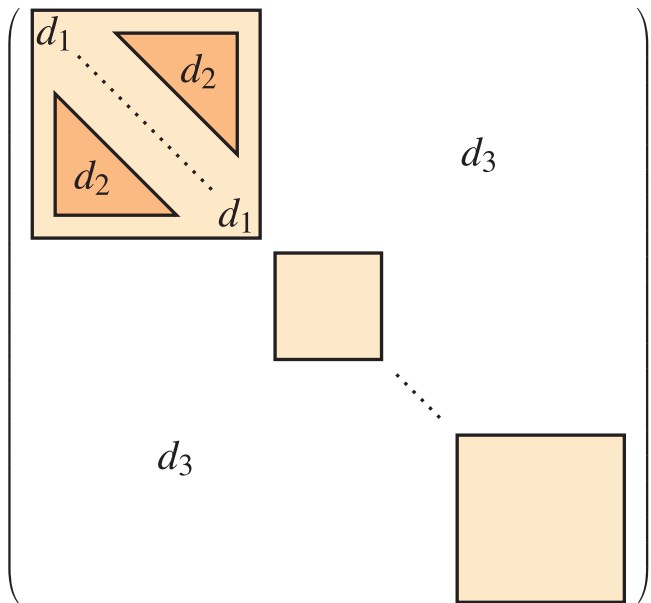

**Figure 2 Probability matrix of the multi-level stochastic block model.**

level structure of any number of levels, but in our experiments we have used 2 levels for the sake of simplicity.

By varying the relation between $d_1$ and $d_2$, we can give different strength to the multi-level community structure. If $d_2$ is close to or equal to $d_1$, the lower level of communities will not be distinguishable and will merge into the upper level. If $d_1$ is instead much larger, then the smaller communities will become more visible. Similarly, if $d_2$ is not significantly larger than $d_3$, the upper level structure will be weak.

We would not consider cases where $d_1 < d_2$ and $d_2 < d_3$, as the resulting structure would be closer to a $k$-partite graph (with $k$ being the number of blocks on the corresponding level) than to a community structure (*i.e.*, blocks would be more connected to each other than to themselves).

## Preferential attachment model

An alternative way to generate benchmark graphs that resemble real networks is the Barabási-Albert preferential attachment model (*Barabási & Albert, 1999*). In this model, new vertices are added successively, and at each addition, a fixed number $m$ of new edges are added connecting the new vertex to the rest of the network, with probabilities of linking to each of the existing vertices proportional to their current degree.

According to the historical account given by *Barabási (2012)*, "*preferential attachment made its first appearance in 1923 in the celebrated urn model of the Hungarian mathematician György Pólya*". The idea (with different naming) reappeared repeatedly over the past century, particularly in the social sciences. A landmark is the *cumulative advantage distribution* of *Price (1976)* that models the citation frequencies of scientific articles, *i.e.*, what we know today as the directed network of citations.

An extension to the preferential attachment model to include communities has been explored in *Hajek & Sankagiri (2019)* and *Jordan (2013)*, which consists on basing the preferential attachment not only on the degree of the vertices, but also on a fitness matrix that depends on their labels (*i.e.*, which block each of the vertices belongs to). This construction can be seen as a weighted preferential attachment, with weights being the affinities between vertex labels. We propose a variation of this preferential attachment block model where new half edges are first randomly assigned a community (with probabilities given by the affinity matrix), from which we then sample the vertex with standard preferential attachment.

The model consist of a sequence of graphs $\{G_t = (V_t, E_t) : t \geq t_0\}$, where $V_t = [t]$ (hence there are $t$ vertices) and $|E_t| = mt$, no self-loops, with the possibility of having parallel edges, and communities $C = (C_1, \dots, C_r)$, with distribution determined by the following parameters:

- $m \geq 1$: fixed number of edges added connecting each new vertex to the graph.
- $r \geq 1$: fixed number of communities.
- $\beta$: $r \times r$ fitness matrix that determines the probability of edges connecting each pair of communities.
- $p = (p_{c_1}, p_{c_2}, \dots, p_{c_r})$: vertex community membership probability distribution.
- $t_0 \geq 1$: initial time (which is also the order of the $G_0$ graph).
- $G_{t_0}$: initial graph

Given graph $G_t$ and community memberships $c = (c_1, \dots, c_t)$, the graph $G_{t+1}$ is generated as follows: A new vertex $t + 1$ is added with community membership sampled from the probability vector $p$, and with $m$ half-edges attached. To obtain the other ends of these half edges, $m$ communities are sampled with replacement with probabilities weighted by $\beta_{c_t,1}, \dots, \beta_{c_t,r}$, and for each of them, a vertex is sampled within the community with preferential attachment (that is, with probabilities proportional to the degree of each vertex). The initial graph $G_{t_0}$ has to be chosen carefully as we will explain below, and further we will show that the resulting network has a scale-free degree distribution.

### Generating the initial graph

The initial graph $G_{t_0}$ is crucial for computing $G_t$ in discrete time, because early vertices have a higher probability to end up with a high degree than later ones due to the nature of the preferential attachment model. To avoid bias with respect to any of the communities, we assign the initial vertices with the same vector of probabilities $p = (p_{c_1}, \dots, p_{c_{|P|}})$ used later when adding new vertices. Then, we sample $t_0 \times m$ edges between them with probabilities proportional to the fitness matrix. In order for $G_{t_0}$ to be able to have enough edges without parallel or self edges, we need $t_0 - 1 > 2m$ (if we have the equality, $G_{t_0}$ will be an $t_0$-clique). It is suggested to use $t_0 = 5m$ to produce a graph that is not too close to a clique, because $G_{t_0}$ is generated with community structure according to a fitness matrix, but the closer it is to a clique, the less this structure matters (at least if we sample without replacement to avoid parallel edges).

## Degree distribution

In this section we will analytically obtain the expected degree distribution of the model. Recall that for the original Barabási-Albert model it is done as follows (shown in *Albert & Barabási, 2002*). Let $k_i$ be the expectation of the degree of node $i$. Then,

$$\frac{dk_i}{dt} = m\frac{k_i}{\sum_{j=1}^{N-1} k_j} \tag{1}$$

The sum in the denominator is known, since the total amount of edges at a certain point is fixed by the model, so $\sum_{j=1}^{N-1} k_j = 2mt - m$, and thus $\frac{dk_i}{dt} = \frac{k_i}{2t-1}$. For large $t$ the $-1$ term can be neglected, and hence we get $\frac{dk_i}{k_i} = \frac{dt}{2t}$. Integrating and taking exponents we get $Ck_i = t^{\frac{1}{2}}$, for some constant $C$. By definition, $k_i(t_i) = m$, and hence $C = \frac{t_i^{\frac{1}{2}}}{m}$. Substituting this value into the last equation, we obtain the desired power law:

$$k_i(t) = m\left(\frac{t}{t_i}\right)^{\frac{1}{2}} \tag{2}$$

For our block model the argument goes as follows. We will assume $k_i$ belongs to the first community to simplify the notation (by symmetry the same principles work on all communities). The rate at which the degree grows is given by

$$\frac{dk_i}{dt} = \left(\sum_{j=1}^{r} p_j\beta_{j1}\right)\frac{k_i}{\sum_{j\in C_1} k_j}m, \tag{3}$$

where the denominator is the expected sum of degrees in community 1:

$$E\left[\sum_{j\in C_1} k_j\right]_t = tm\left(p_1 + \sum_{j=1}^{r} p_j\beta_{j1}\right). \tag{4}$$

Then, Eq. (3) becomes

$$\frac{dk_i}{dt} = \left(\sum_{j=1}^{r} p_j\beta_{j1}\right)\frac{k_i}{t\left(p_1 + \sum_{j=1}^{r} p_j\beta_{j1}\right)} = A\frac{k_i}{t}, \tag{5}$$

where $A = \left(\sum_{j=1}^{r} p_j\beta_{j1}\right)/\left(p_1 + \sum_{j=1}^{r} p_j\beta_{j1}\right)$ is a constant that depends only on the parameters of the model ($p_1,\dots, p_r$ and $\beta$). Then, we integrate the equation:

$$\int \frac{dk_i}{k_i} = \int A\frac{dt}{t} \tag{6}$$

to obtain that $k_i = Ct^A$, for some constant $C$. Now, using the fact that $k_i(t_i) = m$, we obtain the value of the constant as $C = \frac{m}{t_i^A}$, which results in $k_i$ being given by

$$k_i = m\left(\frac{t}{t_i}\right)^A. \tag{7}$$

By an argument similar to one in *Albert & Barabási (2002*, Sec. VII.B) we use Eq. (7) to write the probability that a node has degree $k_i(t)$ smaller than $k$ as

$$P(k_i(t) < k) = P\left(t_i > \frac{tm^{1/A}}{k^{1/A}}\right) = 1 - \frac{tm^{1/A}}{(t + m_0)k^{1/A}}. \tag{8}$$

The last equality obtained from assuming that nodes are added to the network at equal time intervals, and in consequence $P(t_i) = 1/(t + m_0)$. Using Eq. (8) one can readily conclude that the degree distribution of the community $P(k) = \partial P(k_i(t) < k)/\partial k$ is asymptotically given by: $P(k) \sim 2m^{1/A}k^{-\gamma}$, where $\gamma = 1/A + 1$.

## Software

All results in this article have been produced with the *clustAnalytics* R package (*Renedo-Mirambell, 2022*) which contains the needed functionalities for the analysis of clustering algorithms on weighted or unweighted networks, particularly the implementations of the scoring functions, as well as the preferential attachment models with communities. The scripts and data used to perform the experiments are available in Zenodo repository (*Renedo-Mirambell & Arratia, 2023*).

# CLUSTER METRICS ANALYSIS

## Standard SBM network

Using stochastic block models—particularly the *l*-partition model, given the use of the same fixed probabilities for all blocks (cf. "Stochastic Block Model")—we generate a collection of networks with predetermined clusters of varying sizes. The networks are generated as follows: The number of vertices $n$ is fixed, and then, each vertex is assigned to a community with probability $p_{c_1}, p_{c_2}, ..., p_{c_{|P|}}$ (such that $\sum P_i = 1$). Then, this set of probabilities is what will determine the expected sizes of the clusters. Finally, once each vertex is assigned to a community, the edges are generated using the stochastic block model with probabilities $p_{in} = 0.1$ and $p_{out} = 0.001$ (which control the probability of intra and inter community edges, respectively).

To generate $p_{c_1}, p_{c_2}, ..., p_{c_{|P|}}$ we sample $x_1, ..., x_P$ from a power law distribution with $\beta = 1.5$, and then use the probabilities $p_i = \frac{x_i}{\sum_j x_j}$. For this experiment, we have used networks of 300 nodes, with a number of clusters ranging from 5 to 25. For each number of clusters, 1,000 samples have been generated.

Since both the internal and external densities remain constant across all clusters, a strong correlation of a quality metric with cluster size could suggest that it is biased. We can then study the correlation between each quality metric like cluster size (for cluster-wise scores), mean cluster size, and number of clusters relative to graph size (for global scores). Results are shown in Table 3.

Note that by the properties of our model, some of the correlations are to be expected and are a direct consequence of their definitions. This is the case of the cut ratio, internal density or density ratio, which remain nearly constant because they are determined by the values of $p_{in}$ and $p_{out}$, which are constant across all networks. We must remark the very

**Table 3 Pearson correlation table for both global and local scores with respect to size on the SBM.** For the local scores, the first two rows are weighted means, giving a score for the whole network. Note that the last three rows correspond to global scores, so there is no value given for the correlation with local cluster size.

| | Global | | Local size |
|---|---|---|---|
| | #Clusters | Size (Mean) | |
| Internal density | 0.0830 | −0.0610 | −0.0606 |
| Edges inside | −0.6587 | 0.7620 | 0.9027 |
| FOMD | −0.6308 | 0.5951 | 0.8126 |
| Expansion | 0.6855 | −0.7388 | −0.3388 |
| Cut ratio | 0.0095 | −0.0113 | 0.0010 |
| Conductance | 0.9718 | −0.9432 | −0.8164 |
| Norm cut | 0.9682 | −0.9369 | −0.7674 |
| Max ODF | 0.9154 | −0.9320 | −0.8005 |
| Average ODF | 0.9671 | −0.9390 | −0.8016 |
| Density ratio | −0.3079 | 0.2648 | 0.0913 |
| Modularity | −0.3169 | 0.2096 | – |
| Coverage | −0.9234 | 0.8778 | – |
| Global density ratio | −0.0035 | 0.0069 | – |

high correlations of both conductance and normalized cut with the number of clusters. For these two metrics, lower scores indicate better clusterings, so they overwhelmingly favour coarse partitions into few large clusters. We also observe a similarly strong correlation in the case of the coverage metric, but in this case it is a trivial consequence of its definition (coarser partitions will always have equal or better coverage, with the degenerate partition into a single cluster achieving full coverage). In contrast, the modularity only shows very weak correlations to either the number of clusters or their average size.

In the case of the density ratio, we observe that the global version has no correlation to mean size or number of clusters, and that the local version has no correlation with individual cluster size. There is a weak correlation of the weighted mean of the local density ratio with the global properties of the network, which we attribute to the higher likelihood of outliers whenever there is a high number of clusters. Since the score has no lower bound and an upper bound of 1, outliers can have a very small values of the local density ratios, with a great effect on the mean. That is why we suggest using the global density ratio and not the weighted mean of local density ratios when evaluating a network clustering as a whole, and only using the local version when studying and comparing individual clusters.

## Multi-level SBM

We use the multi-level stochastic block model to identify whether a certain metric favours either finer or coarser partitions. Given values of $d_1$ and $d_3$, we can define a parameter $0 \leq \lambda \leq 1$ that will set the strength of the lower level of clustering (if $\lambda = 1$ the upper level dominates, if $\lambda = 0$, the lower level dominates), and then set $d_2 = d_3 + \lambda(d_1 - d_3)$. Upper

level representing coarser partitions than those in lower level, being the latter a refinement of the former.

To evaluate each metric on different configurations of the multi-level community structure, we set a benchmark graph with four communities of 50 vertices on the upper level, each of which splits into two 25 vertex communities on the lower level. We then generate samples across the whole range $[0, 1]$ of values of $\lambda$, with $d_1 = 0.2$ and $d_3 = 0.01$. In general, it is sufficient to choose $d_1$ and $d_3$ different enough so that there is a strong community structure on the upper level, and to have a good range of $d_2$ values that lets us explore their effect on the relation between the two levels of communities.

Figure 3 shows that all scores fail to capture the multi-level nature of the clustering except for modularity and density ratio. The point at which the scores of the lower and upper levels cross on the plot gives us the value of $\lambda$ for which both clusterings are considered equally good by the metric. Then, for values of $\lambda$ smaller than the one attained at the crossing point, the lower level structure is considered preferable, while for larger values, it's the upper level. Then, the value of $\lambda$ at which we find this tipping point characterises the propensity of a metric to favor finer or coarser partitions. In this case, since this occurs for a smaller value of $\lambda$ on the modularity (about 0.15) than the density ratio (a bit over 0.20), we can conclude that the former favors coarser partitions than the later.

As for the rest of the metrics, they always prioritize one level of clustering over the other across all the range of $\lambda$ (that is, the score lines don't cross). Even the conductance and normalized cut, which take into account both internal and external connectivity, fail to give a better score to the lower level when $\lambda$ is zero. Note that when $\lambda = 0$, our model on the upper level is equivalent to an *l-partition* model of 8 blocks which have been clustered arbitrarily joined in pairs, and that still gets a similar or better score than the correct finer partition into the 8 ground truth clusters.

## Multi-level preferential attachment

Here we describe how to set the parameters of the preferential attachment block model defined in "Preferential Attachment Model" to obtain graphs with a multi-level or hierarchical community structure. This is given by the selection of the vector $p$ and the matrix $\beta$. Let $p_{l_1} = (p_1, ..., p_r)$ be the vector of probabilities for the upper level, where $p_i$ is the probability of membership to community $C_i$ (see Fig. 4). Then, similarly to "Multi-level SBM", we want to define a lower level structure that can vary according to a parameter $\lambda$, which will determine which level dominates. We will define $C_{i,1}, ..., C_{i,s_i}$ as the lower level sub-communities of the higher level community $C_i$. Then, to sample the membership of vertices when generating a multi-level preferential attachment graph, we need a vector of probabilities $p_{l_2} = (p_{1,1}, ..., p_{1,s_1}, ..., p_{r,1}, ..., p_{r,s_r})$, where $p_{i,j}$ is the probability of membership to $C_{i,j}$, and such that $\sum_{j=1}^{s_i} p_{i,j} = p_i$ for all $i \in \{1, ..., r\}$, and $\sum_{i=1}^{r} p_i = 1$.

Then, to sample the membership of each vertex on both levels, we simply need to sample its lower level membership according to the probability weights in $p_{l_2}$, which will also induce its upper level membership. As for the affinity matrix (see Fig. 5), it will have

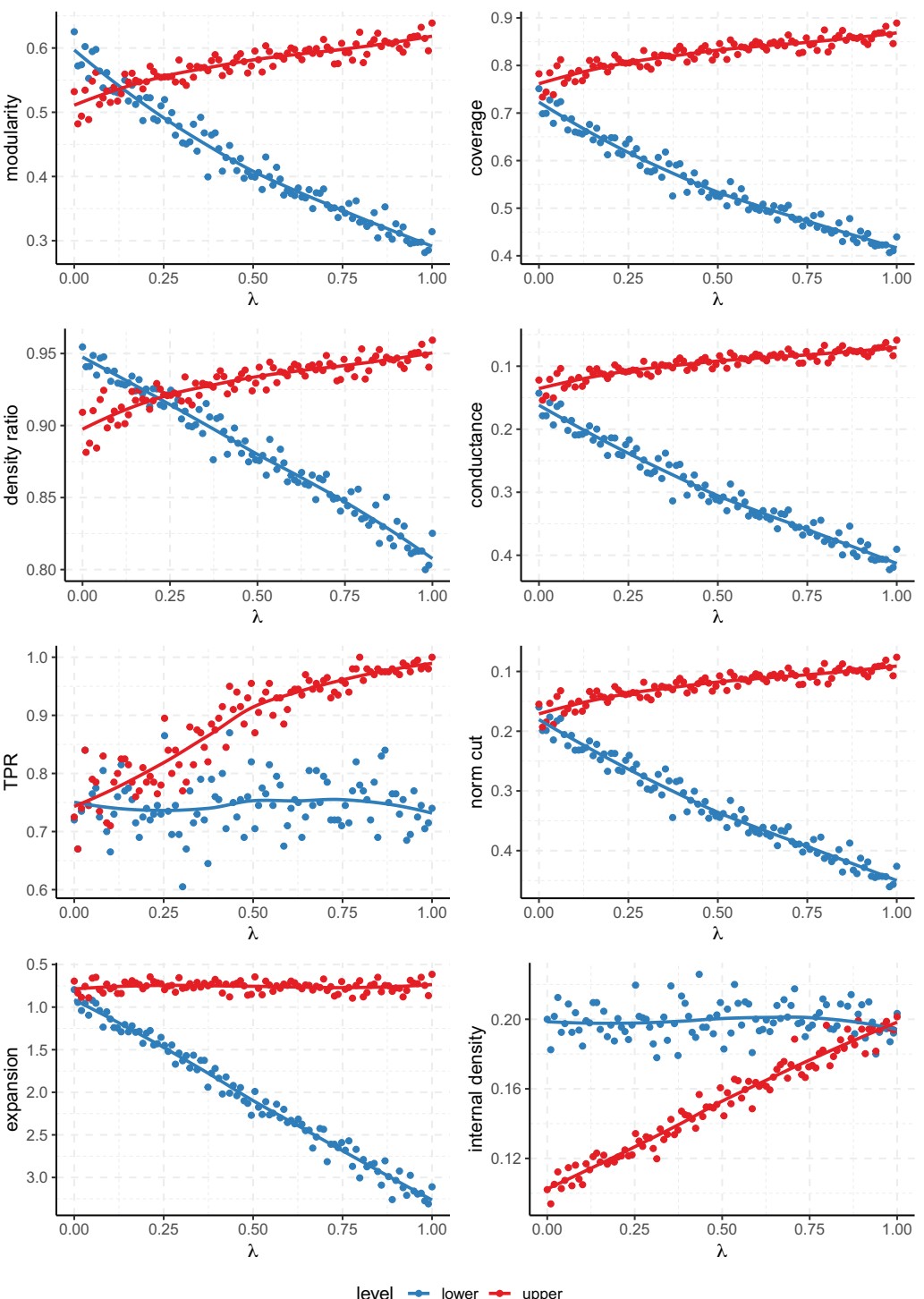

**Figure 3 Values of the quality metrics on each level of clustering on the Multilevel SBM.** $\lambda$ controls the strength of the multi-level community structure. The $y$ axis has been inverted for the scores where lower values are better (conductance, normalized cut and expansion).

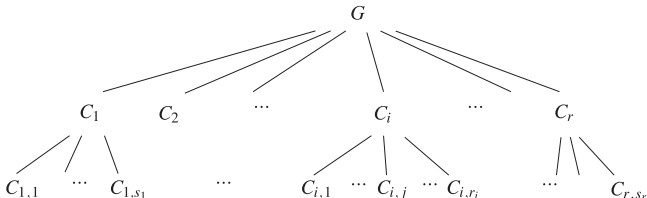

**Figure 4 Diagram of the multi-level preferential attachment graph with community structure following the previously defined notation.**

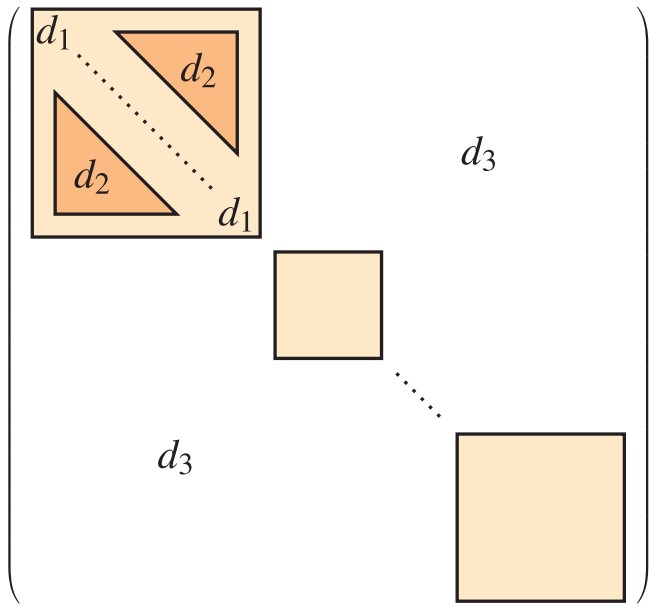

**Figure 5 Affinity matrix of the multi-level block model with preferential attachment.**

the same block structure as the probability matrix of the multi-level stochastic block model in "Multi-level SBM".

Again, we use the parameter $0 \leq \lambda \leq 1$, which controls the values of $d_2$ as follows:

$$d_2 = d_3 + \lambda(d_1 - d_3), \tag{9}$$

and then, following the preferential attachment model we sample edges attached to a new vertex $i$ with probability distribution $(\deg(1)d_{1,i}, \ldots, \deg(i-1)d_{(i-1),i})$. For our experiments, we set $d_1 = 0.2$, $d_3 = 0.01$, and generate samples of $\lambda$ across the whole $[0, 1]$ interval, just as with the multi-level SBM in "Multi-level SBM". $m$ has been set at 4. The results for a network of 300 nodes are shown in Fig. 6.

The results are consistent with those of the multi-level SBM, and again only the modularity and density ratio prioritize either partition depending on the strength of the parameters (the rest always favour the same level of partition). This is seen when the plot lines for the lower and upper level partitions cross, and the value of $\lambda$ at which the lines cross tells us at which degree of relative strength both levels of clustering receive the same

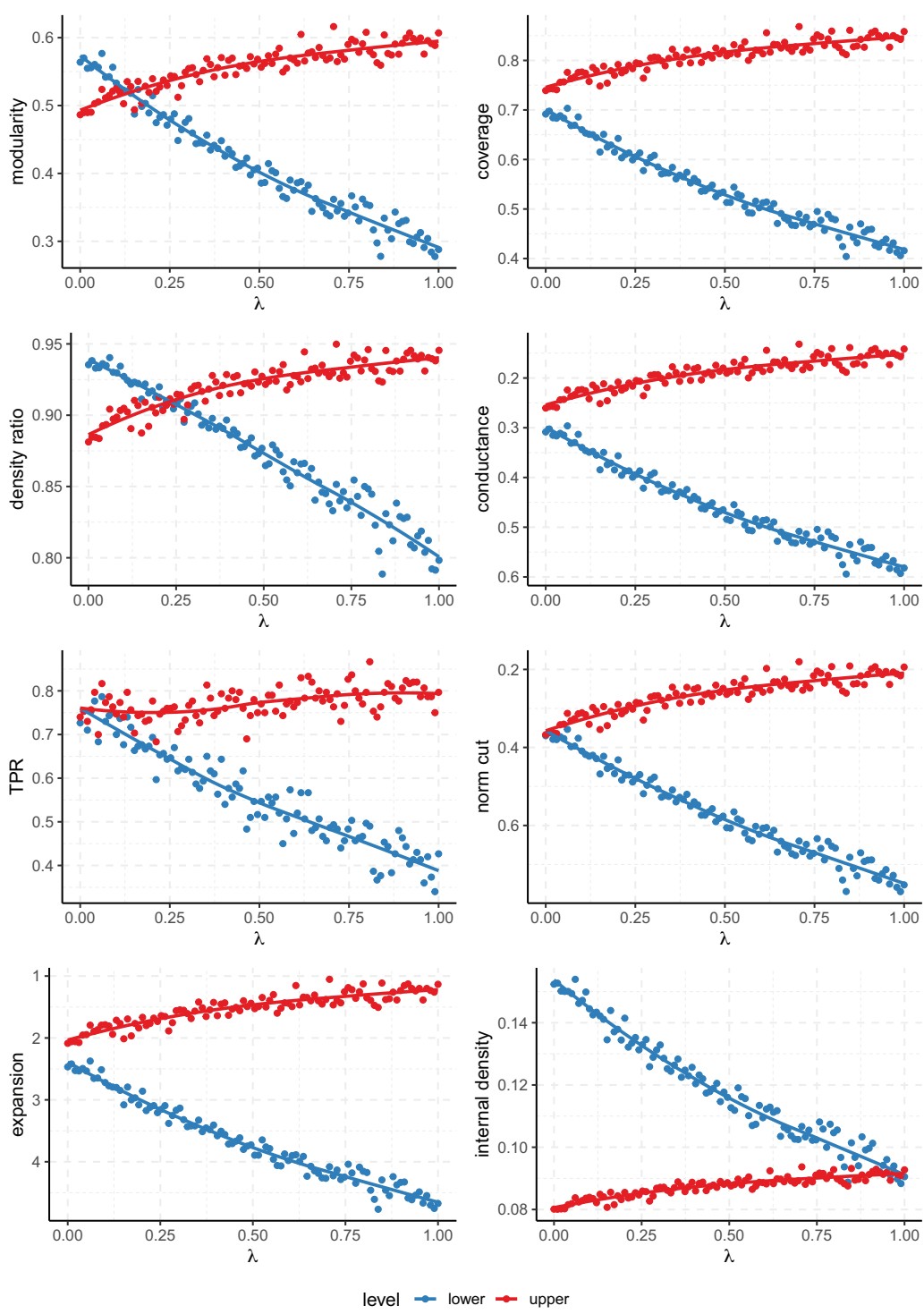

**Figure 6 Values of the quality metrics on each level of clustering for the multi-level preferential attachment model with 300 nodes.** $\lambda$ controls the relative strength of the multi-level community structure. The *y* axis has been inverted for the scores where lower values are better (conductance, normalized cut and expansion).

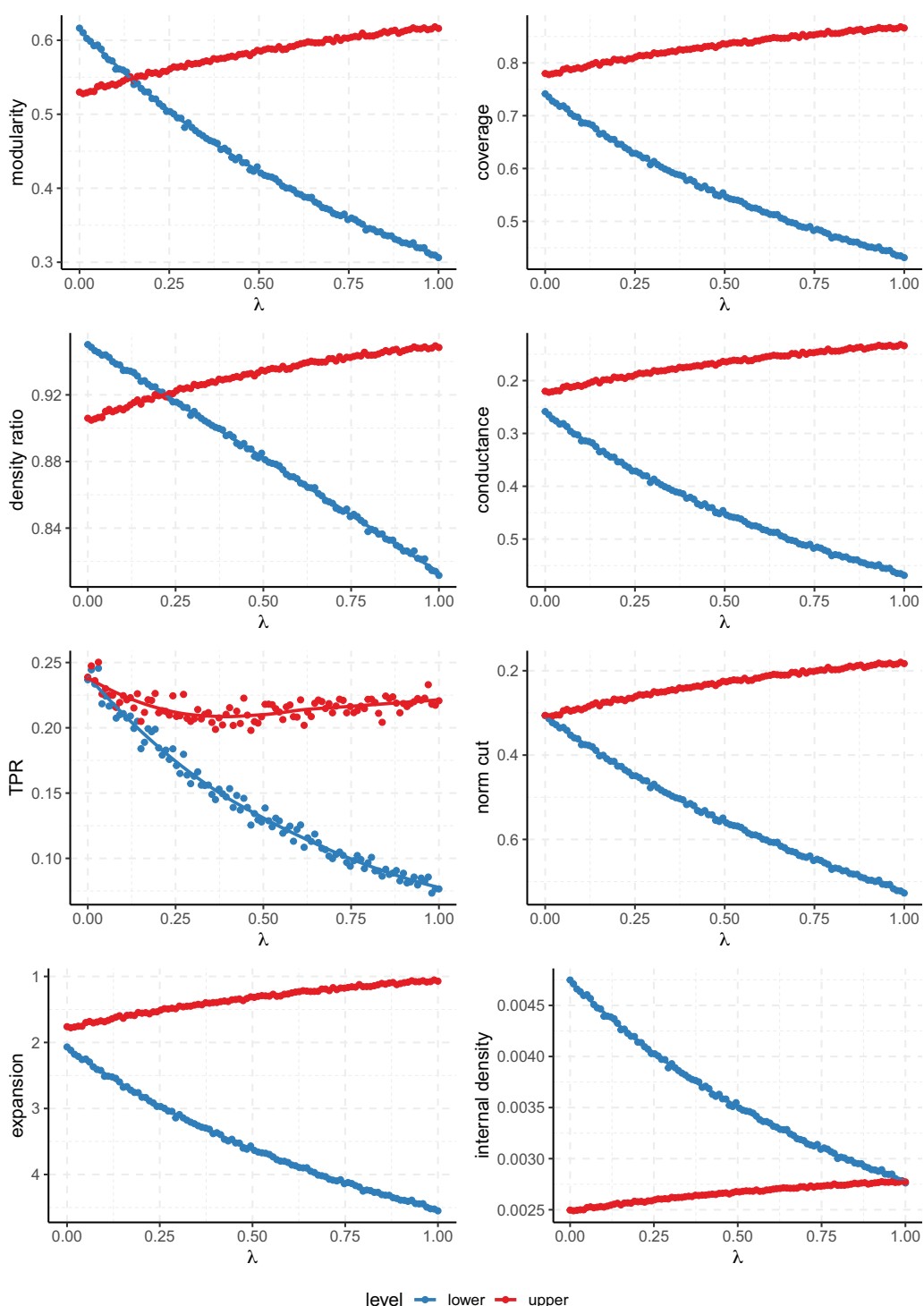

**Figure 7 Values of the quality metrics on each level of clustering for the multi-level preferential attachment model with 10,000 nodes.** $\lambda$ controls the relative strength of the multi-level community structure. The $y$ axis has been inverted for the scores where lower values are better (conductance, normalized cut and expansion).

score. Ultimately, this value of $\lambda$ characterizes to what extent any given score favours fine or coarse partitions.

The same experiment has been repeated on a larger node of 10,000 nodes (see Fig. 7). In this case, there are 8 upper level communities, of which four contain two sub-communities, and two containing both four and eight each. The values of the rest of the parameters have not been changed. As expected, the results are again very similar to both the small preferential attachment and the SBM networks. In this case, the data points deviate less from the fitted curve because on a larger network the empirical scores will be closer to the expected values determined by the parameters of the model.

## CONCLUSIONS

We have observed that most of the considered community metrics are heavily biased with respect to cluster size. While this does not mean that they are useless for cluster quality evaluation, it makes them inadequate for a simplistic approach based on either of them individually. They do however characterize properties that are expected of good clusters, and can complement other methods on a more qualitative analysis. Considering that there isn't a single universal definition of what constitutes a good clustering, being able to evaluate each of these properties separately can be valuable. Also note that these metrics can be particularly useful when comparing partitions of the same number of elements, because in that case the potential of bias related to cluster size is not a concern.

The results of the tests performed on both multi-level models are similar and show that both the modularity and our newly introduced density ratio are capable of evaluating multi-level community structures successfully. When compared among them, though, the modularity favors slightly coarser partitions. Therefore, there are grounds for further analysis of the density ratio metric in future work, such as evaluating it both on well known benchmark graphs and real world networks. It would also be particularly interesting to see how it fares in circumstances were the modularity has limitations, such as when there are strong clusters below its resolution limit.

Additionally, the methods we propose for studying network metrics using multi-level models gave valuable insight and can be of use in to study any new metrics that might be introduced in the future. We strongly suggest the use of metrics that can appropriately detect clusters at different scales when comparing the results of clustering algorithms, because as we have shown, otherwise cluster size has too much influence on the result.

### Funding
The authors received no funding for this work.

### Competing Interests
The authors declare that they have no competing interests.

## Author Contributions

- Martí Renedo-Mirambell conceived and designed the experiments, performed the experiments, analyzed the data, performed the computation work, prepared figures and/or tables, authored or reviewed drafts of the article, and approved the final draft.
- Argimiro Arratia conceived and designed the experiments, analyzed the data, authored or reviewed drafts of the article, and approved the final draft.

## Data Availability

The scripts and data used to perform the experiments are available on Zenodo:

Martí Renedo Mirambell, & Argimiro Arratia Quesada. (2023). Scoring functions bias. Zenodo. https://doi.org/10.5281/zenodo.8100915.

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
