# Peer review of "Identifying bias in network clustering quality metrics"

_PeerJ Computer Science, doi:10.7717/peerj-cs.1523_

## Round 0.1 · original submission · Major Revisions

The reviewers find the work interesting and with potential, although one of them proposes some recommendations to improve the manuscript. It is suggested to provide more specific literature references and address concerns regarding the experimental design, particularly the rationale behind bias and parameter selection. The findings are intriguing but constrained by the small network size and the emphasis on descriptive community detection methods. Enhancing these aspects, as well as tackling resolution issues, would bolster the validity of the findings.

·

Basic reporting

The language could use a bit of proofreading in places to make it more clear and concise throughout.

The literature and background provide enough support and detail to motivate the present work.

The authors provide GitHub for their code as well as an R package, which I believe many will find useful.

The formal results are clear.

Experimental design

The simulation design provided novel additions to the field that I look forward to using myself. I have no issues with the simulation and feel the authors provided a strong enough rationale for the merits of the simulation.

The design and methods were technical but with sufficient information to be understood.

Validity of the findings

Given the R package and GitHub code, I feel that the simulation could be reproduced. The models and simulated data were provided with sufficient justification.

The conclusions did not over state the results and were founded through a solid simulation.

Additional comments

All-in-all, I have no major concerns with this manuscript. The authors did a great job at preparing and executing their study.

Cite this review as

Reviewer 2 ·

Basic reporting

Overall the manuscript reads well aside from some minor issues.

typos and minor comments:
- line 60 and line 64 the references have been repeated, e.g., "Almeida et al Almeida et al"
- section 2.6 starts with the sentencs "The goal is to analytically obtain the expected degree..." --- which goal does this refer to? How does this goal align with the rest of the manuscript?

Literature:
- The idea of planting and detecting two different partitions in a network is also considered in the recent works of Mangold & Roth and Peel & Schaub, which seem complementary to the work of the current manuscript.
- The reference to Lee and Wilkenson on line 69 should be a bit more specific as this is a review of SBMs and variants (similar to Funke & Becker) of which a hierarchical model is one such variant. Perhaps a better reference here would be Peixoto (2014).
- Line 192 refers to the *original* model of Barabasi-Albert, so it would probably be more correct to cite the work of Price.


Funke, T., & Becker, T. (2019). Stochastic block models: A comparison of variants and inference methods. PloS one, 14(4), e0215296.

Mangold, L., & Roth, C. (2023). Generative models for two-ground-truth partitions in networks. arXiv preprint arXiv:2302.02787.

Peel, L., & Schaub, M. T. (2020). Detectability of hierarchical communities in networks. arXiv preprint arXiv:2009.07525.

Peixoto, T. P. (2014). Hierarchical block structures and high-resolution model selection in large networks. Physical Review X, 4(1), 011047.

Price, D. D. S. (1976). A general theory of bibliometric and other cumulative advantage processes. Journal of the American society for Information science, 27(5), 292-306.

Experimental design

The experimental design seems overall fine, but could perhaps be a bit more systematic. The main issue, as I see it, is the reasoning behind the idea of bias. The idea is that because the communities have the same densities, but different sizes (in terms of edges) implies that there are different internal and external degrees. This difference suggests that this is not an issue of bias, but could perhaps be instead an issue of detectability, which can be framed in terms of the difference between internal and external degree relative the the average degree. See Abbe 2017 or Moore 2017 for reviews on this topic. This issue can be addressed either by framing the problem differently and/or extending the experiments.

The choice of some of the parameters for generating the networks seems a bit arbitrary. Not necessarily a problem, but some justification would be good. For instance why choose those values for p_in and p_out and why are they different to the parameters in the hierarchical case (i.e., d_1 and d_3). Also, why change notation? Perhaps more concerning is the size of the networks. With networks this small it is not clear how meaningful the results are or how well the generalize. The manuscript could benefit greatly from exploring a broader range of (and larger) networks. This should be fairly straight-forward given that the networks are synthetically generated.

The experiment is focussed on community quality metrics and specifically on descriptive methods of community detection (see Peixoto 2021) rather than inferential community detection methods. Many of these scores are not designed to be comparable across different networks. For instance, it is known that modularity can have high values even for random graphs with no community structure (Guimera et al).

Not clear about the motivation for using the preferential model, seems that a simpler (and perhaps more consistent) approach would be to generate the desired (heavy-tailed) degree-sequence and use a degree-corrected SBM (Karrer & Newman) --- a hierarchical variant is also availble in Peixoto's graph-tool package.

Abbe, E. (2017). Community detection and stochastic block models: recent developments. The Journal of Machine Learning Research, 18(1), 6446-6531.

Guimera, R., Sales-Pardo, M., & Amaral, L. A. N. (2004). Modularity from fluctuations in random graphs and complex networks. Physical Review E, 70(2), 025101.

Karrer, B., & Newman, M. E. (2011). Stochastic blockmodels and community structure in networks. Physical review E, 83(1), 016107.

Moore, C. (2017). The computer science and physics of community detection: Landscapes, phase transitions, and hardness. arXiv preprint arXiv:1702.00467.

Peixoto, T. P. (2021). Descriptive vs. inferential community detection in networks: pitfalls, myths, and half-truths. arXiv preprint arXiv:2112.00183.

Validity of the findings

Overall the findings are interesting and are valid to an extent, but suffer from the issues described regarding the experimental setup (i.e., size of networks, definition of bias etc.)

One of the findings is that some metrics prefer to arbitrarily group communities. This seems to be a resolution problem (Fortunato & Barthelemy) and indicates that these scores are underfitting.

Fortunato, S., & Barthelemy, M. (2007). Resolution limit in community detection. Proceedings of the national academy of sciences, 104(1), 36-41.

Cite this review as

---

## Round 0.2 · accepted · Accept

The reviewers have found now the paper suitable for publication and I agree with them. Congratulations on your excellent work.

Reviewer 2 ·

Basic reporting

no comment

Experimental design

no comment

Validity of the findings

no comment

Additional comments

The authors have satisfactorily addressed all previous comments.

Cite this review as